# The Impact of the Oil Phase Selection on Physicochemical Properties, Long-Term Stability, In Vitro Performance and Injectability of Curcumin-Loaded PEGylated Nanoemulsions

**DOI:** 10.3390/pharmaceutics14081666

**Published:** 2022-08-10

**Authors:** Jelena B. Đoković, Sotiria Demisli, Sanela M. Savić, Bojan D. Marković, Nebojša D. Cekić, Danijela V. Randjelovic, Jelena R. Mitrović, Dominique Jasmin Lunter, Vassiliki Papadimitriou, Aristotelis Xenakis, Snežana D. Savić

**Affiliations:** 1Department of Pharmaceutical Technology and Cosmetology, Faculty of Pharmacy, University of Belgrade, Vojvode Stepe 450, 11221 Belgrade, Serbia; 2Institute of Chemical Biology, National Hellenic Research Foundation, 11635 Athens, Greece; 3Department of Biochemistry and Biotechnology, University of Thessaly, 41500 Larissa, Greece; 4DCP Hemigal, Tekstilna 97, 16000 Leskovac, Serbia; 5Department of Pharmaceutical Chemistry, Faculty of Pharmacy, University of Belgrade, Vojvode Stepe 450, 11221 Belgrade, Serbia; 6Department of Pharmaceutical Technology and Cosmetology, Faculty of Technology, University of Niš, Bulevar Oslobođenja 124, 16000 Leskovac, Serbia; 7Department of Microelectronic Technologies, Institute of Chemistry, Technology and Metallurgy, University of Belgrade, Njegoševa 12, 11000 Belgrade, Serbia; 8Institut für Pharmazeutische Technologie, Eberhard-Karls Universität, D-72076 Tübingen, Germany

**Keywords:** curcumin, PEGylated nanoemulsions, electron paramagnetic resonance spectroscopy, D-optimal experimental design, fish oil, injectability

## Abstract

A nanotechnology-based approach to drug delivery presents one of the biggest trends in biomedical science that can provide increased active concentration, bioavailability, and safety compared to conventional drug-delivery systems. Nanoemulsions stand out amongst other nanocarriers for being biodegradable, biocompatible, and relatively easy to manufacture. For improved drug-delivery properties, longer circulation for the nanoemulsion droplets should be provided, to allow the active to reach the target site. One of the strategies used for this purpose is PEGylation. The aim of this research was assessing the impact of the oil phase selection, soybean or fish oil mixtures with medium chain triglycerides, on the physicochemical characteristics and injectability of curcumin-loaded PEGylated nanoemulsions. Electron paramagnetic resonance spectroscopy demonstrated the structural impact of the oil phase on the stabilizing layer of nanoemulsions, with a more pronounced stabilizing effect of curcumin observed in the fish oil nanoemulsion compared to the soybean oil one. The design of the experiment study, employed to simultaneously assess the impact of the oil phase, different PEGylated phospholipids and their concentrations, as well as the presence of curcumin, showed that not only the investigated factors alone, but also their interactions, had a significant influence on the critical quality attributes of the PEGylated nanoemulsions. Detailed physicochemical characterization of the NEs found all formulations were appropriate for parenteral administration and remained stable during two years of storage, with the preserved antioxidant activity demonstrated by DPPH and FRAP assays. In vitro release studies showed a more pronounced release of curcumin from the fish oil NEs compared to that from the soybean oil ones. The innovative in vitro injectability assessment, designed to mimic intravenous application, proved that all formulations tested in selected experimental setting could be employed in prospective in vivo studies. Overall, the current study shows the importance of oil phase selection when formulating PEGylated nanoemulsions.

## 1. Introduction

A nanotechnology-based approach to drug delivery presents one of the biggest trends in biomedical science, providing increased active concentration, bioavailability, and safety compared to conventional drug-delivery systems [1]. It is considered a vital resource in delivering high-value, flexible solutions to urgent clinical needs, such as the COVID-19 pandemic [2]. Amongst nanomedicines, nanoemulsions (NEs) stand out for having been present on the market for over 50 years. They were initially used parenterally as the source of calories for patients who could not be orally fed and were later exploited as drug-delivery systems (DDS) for liposoluble drugs, due to their biodegradability, biocompatibility, and relative ease of manufacturing. When administered intravenously, they are metabolized as endogenous chylomicrons or cleared through the mononuclear phagocytic system (MPS) and removed from circulation relatively quickly upon parenteral administration. In order to deliver a drug to the tissues located outside of the MPS, NEs can be surface-modified to change the pharmacokinetic of the droplets upon application or enhance target site selectivity. PEGylation strategy has been used in preparation of parenteral nanoemulsions, in order to prolong the circulation time of the droplets by decreasing the interactions with the MPS. This provides a longer circulation time for the droplets, allowing more opportunity for an active to reach the target site, consequently increasing its concentration on the target site. Various PEGylated phospholipids with different lengths of the PEG chain have been used for this purpose, with the ones with the minimal length of 2000 showing the best in vivo results, for increasing the plasma-circulation time [3].

The selection of the oil phase is paramount to achieve optimal drug loading, adequate physicochemical characteristics of NEs, good shelf-life stability, and desired in vivo fate. Soybean oil (SO) is a staple oil phase in NEs for parenteral nutrition. However, due to its potential of increasing several pathological conditions, it could be beneficial to replace it with oils without inflammatory properties, which can also improve clinical outcomes in certain patients [4]. Fish oil is abundantly comprised of polyunsaturated fatty acids, with DHA (docosahexaenoic acid) and EPA (eicosapentaenoic acid). These fatty acids are recognized as essential nutrients for the human diet, because they can improve or prevent symptoms of cardiovascular disease, cancer, inflammation, arthritis, or allergies [5]. In order to obtain NEs of the best quality, safety, and efficacy profiles, a crucial step is choosing optimal formulation factors and process parameters. To the best of our knowledge, the impact of the oil phase was never considered alongside PEGylated phospholipids, to evaluate their combined impact on the physicochemical characteristics of the PEGylated NEs, in order to help select the best candidates for further research. A design of experiments (DoE) strategy helps to uncover how individual factors, as well as their interactions, impact the critical quality attributes (CQAs) of the NEs, because it allows for their simultaneous consideration. That way, a better control over the final product can be achieved, because the impact of the interactions between the factors cannot be assessed by varying one factor at a time. In this manuscript, we used the DoE method to assess the impact of the type and concentration of the PEGylated phospholipids, oil phase selection, and curcumin presence on the NEs’ CQAs, such as droplet size (Z-ave) and polydispersity index (PDI). Electron Paramagnetic Resonance (EPR) spectroscopy is a technique that provides invaluable information regarding the NE droplet structure, and, in this research, it was used to provide information regarding the oil phase selection’s influence on the NE stabilizing layer and help shed some light on the physicochemical properties and stability of soybean oil and fish oil formulations.

Curcumin is an active ingredient isolated from turmeric (lat. *Curcuma longa*, fam. *Zingiberaceae*) with a plethora of positive effects on the organism, such as anti-inflammatory, antioxidant, antidepressant, antineoplastic, immunomodulatory, and antimicrobial ones [6,7,8,9,10]. Its use has been studied in alleviating many conditions, including depression, arthritis, infections [11,12], and even for improving COVID-19 outcomes [13]. However, its low aqueous solubility and poor bioavailability present a limiting factor in realizing its therapeutic potentials [14] and provide a challenge when developing stable and efficient delivery system, particularly for systemic use. There are several publications investigating NEs as prospective curcumin carriers [15,16,17,18], however, to the best of our knowledge, only a few are dedicated to parenteral delivery [19,20,21], and there are none, bar from our previous research [22] dedicated to the use of PEGyaltion as a way of improving curcumin’s poor physicochemical characteristics and obtaining a product that has characteristics adequate for parenteral drug delivery system, which include good long-term stability.

The aim of this research was to elucidate the impact of the oil phase selection on physicochemical properties, long-term stability, in vitro performance, and injectability/applicability of curcumin-loaded PEGylated nanoemulsions, using the EPR and DoE tools in assessment.

## 2. Materials and Methods

### 2.1. Materials

PEGylated phospholipids (PEG-PLs)—PEG2000-DSPE (N-(Carbonyl-methoxypolyethylenglycol-2000)-1,2-distearoyl-sn-glycero-3-phosphoethanolamine, sodium salt)—purely plant-derived and synthetic raw material (1,2-distearoyl-sn-glycero-3-phosphoethanolamine DSPE and polyethylene glycol) with 95–105% of MPEG2000-PE, stearic acid purity not lower than 98%, and PEG-5000-DPPE (N-(Carbonyl-methoxypolyethylenglycol-5000)-1,2-dipalmitoyl-sn-glycero-3-phosphoethanolamine, sodium salt)—produced from vegetable and synthetic materials, i.e., natural sn-glycero-3-phosphocholine (GPC, derived from soybean), vegetable fatty acids, and synthetic polyethylene glycol, with not less than 96% of the phospholipids, and the purity of palmitic acid not less than 98%, soybean oil (Lipoid Purified Soybean Oil 700), sodium oleate (Lipoid Sodium Oleate B), and soybean lecithin (Lipoid S 75, with 70% of the phospholipids) were purchased from Lipoid GmbH (Ludwigshafen, Germany). Curcumin((E,E)-1,7-bis(4-Hydroxy-3-methoxyphenyl)-1,6-heptadiene-3,5-dione), fish oil, polysorbate 80 (polyoxyethylensorbitanmonooleate), benzyl alcohol (BA), butylated hydroxytoluene (BHT), 2,2-Diphenyl-1-picrylhydrazyl (DPPH), 2,4,6-Tris(2-pyridyl)-s-triazine (TPTZ), iron (III) chloride (FeCl_3_), 5-, 12- and 16-Doxyl stearic acid, and hydrochloric acid were obtained from Sigma-Aldrich Co (St. Louis, MO, USA). Glycerol was provided by Merck KGaA (Darmstadt, Germany), and medium chain triglycerides (MCT) were purchased from Fagron GmbH & KG (Barsbüttel, Germany). Water used for the preparation of formulations as well as for analyses was ultra-pure and obtained via a GenPure apparatus (TKA Wasseranfbereitungssysteme GmbH, Neiderelbert, Germany). All other chemicals and reagents were of pharmaceutical or HPLC grade and were used without further purification.

### 2.2. Methods

#### 2.2.1. Solubility of Curcumin in Oils

Shake flask method was used to determine solubility of curcumin in soybean and fish oil as well as in mixtures of MCT and these oils at 4:1 (*w*/*w*) ratio. Curcumin was added in excess to 5 g of the oils or their mixtures, which were left to stir at 250 rpm on orbital shaker (IKA KS 260 basic, IKA Werke GmbH & Co. KG, Esslingen am Neckar, Germany) for 24 h at 25 ± 2 °C, shielded from light. They were subsequently centrifuged (Centrifuge MPW-56, MPW Med. Instruments, Warsaw, Poland) for 30 min at 5000 rpm, and the concentration of curcumin was determined by diluting the supernatant aliquots in methanol and measuring the absorbance of the resulting solutions at 425 nm on spectrophotometer (Evolution 300, Thermo Fisher Scientific, Cambridge, UK).

#### 2.2.2. Preparation of Nanoemulsions

Nanoemulsions were prepared using hot, high-pressure homogenization technique. In brief, the components of the oil (soybean/fish oil, medium chain triglycerides, soybean lecithin, benzyl alcohol, and butylated hydroxytoluene) and aqueous phases (glycerol, polysorbate, sodium oleate, and highly purified water) were mixed separately and heated on the magnetic stirrer at 50 °C. Hydrochloric acid (0.1 M solution) was used to adjust the pH of the aqueous phase below 7. In the case of curcumin-loaded NEs, curcumin was firstly dissolved in benzyl alcohol then added to the rest of the oil phase, after the lecithin was dissolved. PEG2000-DSPE and PEG5000-DPPE were poured to the oil and aqueous phase, respectively at 0.1%/0.3%/0.6% concentrations. The full composition of the formulations is given in Table 1. Aqueous phase was added to the oil phase and mixed for 1 min at 11,000 rpm using rotor-stator homogenizer (IKA Ultra-Turrax T25 digital, IKA-Werke GmbH & Co. KG, Staufen, Germany), in order to prepare coarse emulsion, which was further processed by high-pressure homogenizer (EmulsiFlex-C3, Avestin Inc., Ottawa, ON, Canada) for 10 discontinuous cycles at 800 bar at 50 °C.

#### 2.2.3. Electron Paramagnetic Resonance (EPR) Spectroscopy Measurements

The interfacial properties of the formulated non-PEGylated Nes in the absence and presence of curcumin were studied by Electron Paramagnetic Resonance (EPR) Spectroscopy (EPR), using the spin-probing technique. This method measures the absorption of microwaves by paramagnetic centers with one or more unpaired electrons. In the present study, three different amphiphilic fatty acid derivatives labeled at different positions of the aliphatic chain, namely 5-, 12-, and 16-doxyl stearic acids, were utilized to probe membrane dynamics at different depths. This method is crucial for understanding the dynamics of the surfactants’ monolayer and the location of the encapsulated compound.

EPR measurements were performed with an EMX EPR spectrometer (Bruker BioSpin GmbH, Rheinstetten, Germany) operating at the X-Band (9.8 GHz) using a quartz, flat, aqueous sample cell (Wilmad-LabGlass, Cortecnet Europe, Voisins-Le-Bretonneux, France). Instrument settings were center field 0.348 T, scan range 0.01 T, receiver gain 5.64 × 10^4^, time constant 10.24 ms, conversion time 5 ms, and modulation amplitude 0.4 mT for 5- and 16-DSA. To obtain the spectra of 12-DSA, receiver gain and conversion time were adjusted to 5.64 × 10^3^ and 5.12, respectively. Sample preparation for EPR measurements was as follows: Initially, stock solutions of the spin probes were prepared in absolute ethanol at a concentration of 1 mM. Then, 15 μL of the stock solution were added to Eppendorf tubes. After ethanol was evaporated, 1 mL of the nanoemulsions was added to each Eppendorf and kept for 1 day at ambient temperature to allow probe solubilization at the surfactants’ layer. The final concentration of the spin-probes in the nanoemulsions was 0.015 mM.

The Bruker WinEPR acquisition and processing program (Bruker BioSpin GmbH, Ettlingen, Germany) was used for data collection and analysis. EPR spectra were analyzed in terms of rotational correlation time (τR), order parameter (S), and isotropic hyperfine coupling constant (αN).

#### 2.2.4. Experimental Design

Experimental design strategy was employed in order to assess the impact of the oil phase selection and the presence of curcumin on the droplet size of PEGylated nanoemulsions. In that vein, four independent variables were chosen for experimental design analysis and their levels were: PEG-PL type [A]—PEG2000-DSPE/PEG5000-DPPE; PEG-PL concentration [B]—0.1%/0.3%/0.6%; oil type [C]—fish oil/soybean oil; and the presence of curcumin [D]—no (absence)/yes (presence), while the droplet size (Z-ave) and polydispersity index (PDI) were chosen as response variable, as they were predicted to be responsive enough. The selection of PEGylated phospholipid types and their concentrations was based on the literature overview and previous trials. The results obtained for each response were statistically evaluated using Design-Expert software v. 9.0.1 trial (Stat-Ease Inc., Minneapolis, MN, USA). The first-order polynomial model was applied in order to fit the experimental data, based on the significant model terms (*p* < 0.05), multiple correlation coefficient (R^2^), adjusted multiple correlation coefficient (adjusted R^2^), non-significant lack of fit, and adequate precision value, as provided by Design-Expert software.

#### 2.2.5. Size Measurements

The droplet size of the formulations was assessed by dynamic light scattering (DLS) using ZetasizerNano ZS90 (Malvern Instruments Ltd., Worcestershire, UK) and presented as the mean droplet size (intensity weighted mean diameter, Z-average diameter, Z-ave) and droplet-size distribution (PDI). The samples were prepared by dilution with ultra-pure water in 1:500 (*v*/*v*) ratio, and the measurements were performed at 25 °C at a fixed scattering angle of 90° using a He-Ne laser at 633 nm.

DLS measurements were supplemented with laser diffraction (LD) technique, in order to determine the potential presence of larger droplets. LD measurements were performed using Malvern Mastersizer 2000 (Malvern Instruments Ltd., Worcestershire, UK), and volume-weighted diameters d (0.5), d (0.9), and D [4,3] were used as sizing parameters.

#### 2.2.6. Polarization Microscopy

Polarization microscopy analysis was conducted using Carl Zeiss ApoTome Imager Z1 microscope (Zeiss, Göttingen, Germany), equipped with the AxioCam 105 camera and Zen Imaging software, in order to detect the presence of any larger droplets or curcumin crystals in the undiluted samples after two years of storage. A drop of the nanoemulsion formulation was placed on the microscope glass slide and captured under 400× magnifications.

#### 2.2.7. Atomic Force Microscopy

For the atomic force microscopy, NE sample (diluted in ultra-pure water in 1:1000, *v*/*v* ratio) was placed onto mica plate (Highest Grade V1 AFM Mica Discs, Ted Pella Inc., Redding, CA, USA) and dried under vacuum to remove the excess water. NTEGRA prima atomic force microscope (NT-MDT) was used to inspect the morphology, shape, size, and distribution of the NE droplets. The intermittent-contact AFM mode was used, due to the samples’ fragility. NT-MDT NSGO1 rectangular silicon cantilevers with Au-reflective film were used for this purpose. Nominal resonant frequency of these cantilevers is 150 kHz, while nominal force constant is 5.1 N/m. Image Analysis 2.2.0 (NT-MDT) software was used to process the obtained data.

#### 2.2.8. Zeta Potential

Zeta potential of the samples was measured upon dilution in the 50 µS/cm sodium chloride solution in 1:500 (*v*/*v*) ratio using ZetasizerNano ZS90 (Malvern Instruments Ltd., Worcestershire, UK).

#### 2.2.9. pH and Conductivity

The pH and conductivity values of the samples were measured by direct immersion of the pH meter and conductometer electrodes into the NE samples.

#### 2.2.10. Drug Content and Encapsulation Efficacy

The content of curcumin in nanoemulsions was ascertained by dissolving them in methanol (1:1000, *v*/*v* ratio), both initially after preparation and after two years of storage, to determine the ability of the formulations to preserve the long-term stability of curcumin. These samples were prepared in triplicate. The encapsulation efficacy (EE) was assessed using the Amicon Ultra-4; NMWL 10 kDa filter units (Merck Millipore, Burlington, MA, USA). Briefly, 2 mL of the formulation was placed in the tube and centrifuged at 2000 rcf for 90 min. The filtrate was mixed with the equal volume of methanol and analyzed for curcumin content. The encapsulation efficacy was determined using the equation: %EE = ((A_formulation_ − A_filtrate_)/A_formulation_) × 100, where A_formulation_ represents the content of curcumin in the NE sample, and A_filtrate_ stands for the content of curcumin in the filtrate. These experiments were performed in duplicate. The content of curcumin in all samples was determined using the LC-MS/MS method, previously described elsewhere [22,23].

#### 2.2.11. Antioxidant Activity

Antioxidant activity of the formulations was assessed through two complementary assays: DPPH and FRAP. The antioxidant potential of the formulations containing two different oils (soybean or fish oil) was compared to that of pure curcumin, in order to evaluate the contribution of the formulation to the overall antioxidant effect.

DPPH assay

The test was performed by adding 0.4 mL of curcumin dilutions in isopropanol into 3.6 mL of 0.1 mM DPPH free radical isopropanol solution, resulting in final curcumin concentrations of 0.00188, 0.00375, 0.00563, 0.00750, 0.01125, and 0.01313 mg/mL, covering absorbance values between approximately 0.3 and 0.8. Each concentration was prepared in triplicate. The absorbance was recorded at 517 nm using UV/VIS spectrophotometer (Evolution 300, Thermo Fisher Scientific, Cambridge, UK), after incubation in the dark, at room temperature for 30 min. The control sample was prepared using isopropanol instead of curcumin dilutions. The free radical scavenging potential was expressed as inhibition percentage and calculated using the equation I = [(Ac × As)/Ac]/100; where I stands for inhibition percentage, Ac for the absorbance of the control sample, and As for the absorbance of the test sample. The calculated inhibition percentage was plotted against sample concentration, and IC50 values (concentration that causes 50% of inhibition or scavenging activity) were determined by linear-regression analysis. Free scavenging activity of the Nes (with soybean oil: CS, CS21, CS51, or fish oil: CF, CF21 and CF51) was tested after dilution with highly purified water, to obtain the same concentration of curcumin as in the isopropanol solutions, followed by the abovementioned procedure.

FRAP assay

In short, 100 µL of the NE (with soybean oil: CS, CS21, CS51, or corresponding fish oil formulations CF, CF21, and CF51) was diluted in water, mixed with 3 mL of freshly prepared FRAP reagent (25 mL of 300 mM acetate buffer (pH 3.6); 2.5 mL of 10 mM TPTZ (2, 4, 6-tripyridyls-triazine) solution in 40 mM HCl; 2.5 mL of 20 mM FeCl_3_ · 6 H_2_O solution in purified water), and left in the dark to incubate for 30 min, at controlled temperature of 37 °C. Afterwards, the absorbance was measured at 593 nm via UV/VIS spectrophotometer (Evolution 300, Thermo Fisher Scientific, Cambridge, UK). The blanks were prepared with the corresponding placebo formulations. The same test was performed using solutions of curcumin in methanol. Antioxidant potential, expressed as FRAP value, was calculated based on the calibration curve obtained by measuring the absorbance of mixtures of 100 µL of FeSO_4_ · 7 H_2_O standard solutions (concentrations ranging from 25 to 1200 µmol/L) and 3 mL of FRAP reagent and presented as mmol Fe^2+^/g of dry matter. The tested concentration of curcumin (0.0375 mg/mL) was chosen based on the absorbance intensity.

#### 2.2.12. Stability Study

Droplet size, zeta potential, pH, conductivity, curcumin content, and antioxidant potential (DPPH and FRAP assays) were assessed initially and after two years of storage.

#### 2.2.13. In Vitro Release

The release of curcumin from the NEs was studied through the dialysis bag technique, using D-9527 Sigma cellulose membrane dialysis tubing (molecular weight cut-off 12,000), and was overnight-soaked in the dissolution medium—a mixture of highly purified water and ethanol (1:1, *v*/*v*). The dialysis tubing/bags were filled with 2 mL of NE sample (donor phase) and sealed with dialysis-tubing closers. The tubing was placed in a flask containing 200 mL of dissolution medium and covered with aluminum foil to prevent the evaporation of ethanol and light induced degradation of curcumin. The test was performed in ES-20 orbital shaker–incubator (Biosan SIA., Riga, Latvia), at 37 °C. After 5, 20, 40, 60, 120, and 240 min, the sample was retrieved from the flask (acceptor phase) and replaced with equal amount of the release medium to maintain the sink conditions. The samples were further analyzed for curcumin content by abovementioned LC-MS/MS (please see Section 2.2.10). The analysis was performed in triplicate. The release pattern of curcumin from selected formulations was calculated by plotting cumulative percentage drug release vs. time, while the release kinetics of investigated NEs were evaluated by fitting the experimentally obtained release data through several mathematical models (zero-order, first-order, Higuchi, Baker–Lonsdale, Korsmeyer–Peppas, and Hixson–Crowell), by employing DDSolver add-in program for the Microsoft Excel application.

#### 2.2.14. Viscosity and Injectability

Rheological analysis was performed using MCR 302 air-bearing rheometer (Anton Paar, Graz, Austria) equipped with coaxial cylinders system (CC27 measuring bob with C-PTD 180/Air) with sheer rate range of 0.1–100 s^−1^ at 20 °C.

The injectability of the NEs was defined as the force needed to displace the plunger of the syringe in a function of the extruded volume (mL). About 10 mL of the formulation was loaded into the 10 mL syringe and extruded through the 25 G scalp vein infusion set (Romed, Wilnis, The Netherlands) into the circulating-blood-mimicking solution, to observe the behavior of the NEs in the prospective intravenous administration. The NEs were extruded at 1 mm/s crosshead speed of the loading cell of the Texture analyzer (EZ-LX Compact Table-Top Testing Machine, Shimadzu, Japan), with the TrapeziumX software version 1.5 used for data collection and analysis.

#### 2.2.15. Statistical Analysis

Statistical analysis was performed using the IBM SPSS Statistics software (v.23) (Chicago, IL, USA). The results were presented as mean values of observed parameter ± SD, whenever possible. The differences in physicochemical parameters, measured upon preparation and after two years of storage, as well as the differences between the amounts of curcumin released from non-PEGylated formulation, were studied using Student’s *t*-test. For the analysis of the results obtained from in vitro release testing of the PEGylated formulations, one-way ANOVA, followed by Tukey HSD post hoc test, was used. When the assumptions of ANOVA were not met (the variances of the groups were not equal), a nonparametric Kruskal–Wallis test (one-way ANOVA by ranks) was performed, together with the nonparametric Mann–Whitney U test, for pairwise group comparisons, in the cases where significant difference was observed. The *p* value of 0.05 was taken as significant.

## 3. Results

The aim of solubility testing was to determine whether or not the use of fish oil would impact drug loading. The solubility of curcumin in soybean and fish oil was 0.7551 mg/mL and 0.5715 mg/mL, respectively. In the soybean/fish oil and MCT mixtures (1:4, *w*/*w*) ratio, the solubility of curcumin was 1.3870 mg/mL for the soybean mixture and 1.4526 mg/mL for the fish oil mixture.

EPR spectroscopy technique was used to elucidate the structural implications of the oil phase selection. Three-line EPR spectra of unequal heights and widths characteristic for doxyl derivatives, when located in membranes, were obtained for non-PEGylated samples, both curcumin-loaded –CS and CF as well as placebos PS and PF. As can be seen below, Figure 1a,b depict the experimental spectra of the three probes (5-, 12-, and 16-DSA) in the absence and presence of curcumin in NEs with soybean and fish oil used as the oil phase: PS—placebo with soybean oil; PF placebo with fish oil; CS—curcumin-loaded soybean oil; CF—curcumin-loaded fish oil NE, respectively. From the analysis of these, spectra rotational correlation time (τR), order parameter (S), and isotropic hyperfine coupling constant (αN) were calculated, providing information about the dynamics of the surfactants’ monolayer and the localization of curcumin. [24,25]. The results of the EPR study are shown in Table 2. Detailed equations and the physical meaning of those parameters have been reported in previous studies [26,27]. Irrespective of the oil type or the presence of curcumin, the τR and the S values decreased in the following order: 5-DSA, 12-DSA, and 16-DSA (Table 2). EPR spectra in nanoemulsions containing soybean oil (Figure 1a), and the values obtained from the EPR spectral characteristics (Table 2), demonstrated that the addition of curcumin caused changes in the stabilizing layer, which was reflected in the τR values, which decreased from 2.18 ± 0.60 ns to 1.66 ± 0.61 ns for 5-DSA. The other two spin-probes do not exhibit any significant changes concerning the values of S and αN and only a small decrease in τR (Table 2). On the other hand, in the case of fish oil nanoemulsions, upon the addition of curcumin, an increase in τR for all three spin-probes can be observed (Table 2), particularly for 5-DSA, where changes in the order parameter and the hyperfine constant were also significant. Given that the EPR analysis provides qualitative data regarding the localization of curcumin inside the droplets, an additional encapsulation-efficacy study was performed to assess the amount of curcumin loaded in the NE droplets. The amount of curcumin found in the filtrate was below the quantification limit, meaning the encapsulation efficacy was >99% for both CS and CF samples.

The goal of the DoE was to find out how the oil phase selection, together with the presence of curcumin and the type and concentration of the PEGylated phospholipids, impact the chosen CQAs of NEs, i.e., droplet size and size distribution. Concordant with the DoE matrix (Table 3), 25 different PEGylated NEs samples were prepared. The mean droplet size of all of the prepared formulations was between 95 and 108 nm, and the polydispersity index values were below 0.2, indicating suitability for parenteral administration. The D-optimal experimental design strategy resulted in proposed 25 runs [15 (model points) + 5 (to estimate lack of fit) + 5 (replicates)], which were then preformed at random. The effects of the investigated factors (PEG-PL type and concentration; oil type and the presence of curcumin) and their interactions on dependent variables were calculated.

All of the individual factors and interactions that were deemed statistically insignificant for the responses (*p* > 0.05) were excluded from the Z-ave and the PDI models, unless model terms were required to support hierarchy resulting in raising the reduced factorial models. The equations for Z-ave (1) and PDI (2) in terms of the coded factors are given below:Z-ave = +100.83 + 0.66 A + 0.34 B(1) − 0.07 B(2) − 0.80 C − 1.30 D − 0.045 AB(1) − 0.24 AB (2) − 0.15 B(1)D + 0.091 B(2)D(1)
PDI = +0.14 − 0.008 A + 0.002 B(1) 0.002 B(2) − 0.004 C − 0.003 D − 0.008 AB(1) − 0.003 AB(2) − 0.014 AD + 0.001 B(1)C − 0.003 B(2)C − 0.003 B(1)D + 0.002 B(2)D(2)

The ANOVA analysis demonstrated that the models generated for Z-ave and PDI were significant (*p* ˂ 0.05) with the model F values of 15.40 and 14.76, respectively, and the non-significant lack of fit relative to the pure error. The R^2^, adjusted R^2^, and the adequate precision values of the proposed models (Z-ave: 0.9023, 0.8437, and 12.471; PDI: 0.9366, 0.8737, and 12.367, respectively), indicated that proposed models adequately described the two responses and, therefore, could be used to navigate in the design space.

Appendix A depicts the influence of different factors and their interactions on the mean droplet size (Z-ave) and polydispersity index (PDI). Coefficient values indicated the magnitude of impact, while the sign showed the direction of the model term influence—e.g., the negative sign represented antagonistic effect of the adjoining factor. It can be observed (Appendix A) that all of the investigated factors significantly influenced Z-ave (*p* < 0.05) in the following order: the presence of curcumin [D] > oil type [C] > PEG-PL type [A] > PEG-PL concentration [B]. The negative sign in front of the model drug signified the addition of curcumin resulted in smaller droplets. The oil phase selection had an antagonistic effect on Z-ave, meaning fish oil NEs (CF21, CF23, CF26, CF51, CF53, and CF56) had smaller droplets compared to the soybean oil ones (CS21, CS23, CS26, CS51, CS53, and CS56). The PEGylation also affected the droplet size, with both the PEGylated phospholipid type and concentration having a positive effect on Z-ave, meaning that smaller droplets were obtained using PEG2000-DSPE compared to PEG5000-DPPE, and the increase in either of their concentrations led to the formation of bigger droplets.

Although individual factors had a more significant role on the Z-ave values (Appendix A), the two factor interactions (PEG-PL type/PEG-PL concentration—AB; PEG-PL concentration/ presence of curcumin—BD) were also shown to influence the nanoemulsion droplet size at a statistically significant level (*p* < 0.05). When analyzing the AB interaction, it could be observed that the droplet sizes were similar when using either 0.1% or 0.6% of the PEG2000-DSPE compared to the same concentration of the PEG5000-DPPE. On the other hand, the use of 0.3% of the PEG5000-DPPE led to bigger droplets then with PEG2000-DSPE, regardless of the oil phase composition or the presence of curcumin (Figure 2). The BD-interaction analysis revealed that the presence of curcumin led to smaller droplets being formed for both of the oil phases and the PEGylated phospholipid types, but this effect was more pronounced with 0.3% or 0.6% of the PEG-PLs.

Out of the individual factors, only the PEG-PL type [A] and the PEG-PL concentration [B] had a statistically significant impact on the PDI (*p* < 0.05), with the PEG PL type being more pronounced (Appendix A). The antagonistic effect of the PEGylated phospholipid type indicated that a smaller PDI could be obtained when using PEG5000-DPPE compared to PEG2000-DSPE, while the agonistic effect of the PEG PL concentration indicated higher concentrations caused increased polydispersity index values.

Contrary to the droplet size, PDI was predominantly influenced by the interactions between the factors (PEG-PL type/ presence of curcumin—AD; PEG-PL type/PEG-PL concentration—AB; PEG-PL concentration/oil type—BC; PEG-PL concentration/ presence of curcumin—BD). The AD-interaction analysis (Figure 3) indicated that when using 0.1% of either PEGylated phospholipid, the presence of curcumin resulted in higher PDI values compared to the placebo formulations, with lower values obtained with PEG5000-DPPE compared to PEG2000-DSPE. When 0.3% of the PEG2000-DSPE was used, similar but lower PDI values could be observed for the placebo formulations, as opposed to the curcumin-loaded ones, contrary to the NEs with PEG5000-DPPE, where lower PDI values were obtained for the curcumin-loaded formulations. With 0.6% of the PEGylated phospholipids, higher PDI values were observed for the placebo formulations compared to the curcumin-loaded ones, with lower values gained when using PEG5000-DPPE compared to PEG2000-DSPE, regardless of the oil type. The AB interaction (Figure 3) indicated that the lowest PDI for the placebo formulations could be obtained with 0.1% of PEG-PL, regardless of the oil phase selection, whereas the addition of curcumin led to similar PDI values being obtained when using 0.1% or 0.6% of the PEG2000-DSPE in both oil phases. A similar trend with lower PDI values could be observed with PEG5000-DPPE. The BC interaction (Figure 3) showed that the use of fish oil instead of soybean oil decreased PDI values for formulations with 0.1% and 0.6% of either PEG2000-DSPE or PEG5000-DPPE, while in the formulations with 0.3% of the PEGylated-PLs, it led to a PDI increase. The BD-interaction analysis demonstrated the increase in the PEGylated phospholipid concentration led to increased PDI values for formulations with PEG2000-DSPE and fish oil, with a more pronounced effect on the placebo formulations. In the case of soybean oil, the use of 0.3% of the PEG2000-DSPE led to a PDI decrease compared to the other two concentrations. With PEG5000-DPPE formulations, the 0.3% concentration led to the highest PDI values in fish oil NEs, regardless of the presence of curcumin, while in the soybean oil ones the increase in PEGylated phospholipid concentration led to a decrease in PDI values for curcumin-loaded NEs and an increase for the placebo ones (Figure 3).

The initial values of physicochemical parameters for all formulations (Table 1) were NE droplet size (<200 nm), polydispersity index (<0.2), zeta potential (about −30 mV to—40 mV), pH (about 7.4), and conductivity values (about 180 µS/cm) (Table 4, Table 5, Table 6, Table 7 and Table 8), indicating suitability for parenteral administration. The atomic force microscopy demonstrated the presence of small spherical droplets, about 100 nm in diameter, in CF formulation, while the addition of 0.1% of PEG2000-DSPE (CF21) seemed to impact the characteristics of the formulations during drying, so only larger droplets, around 300 nm, could be observed (Figure 4). In the soybean oil NEs, both PEGylated and not, CS and CS21, respectively, were similarly sized, and shaped droplets were detected (Figure 5).

The oil phase selection had a more pronounced effect on the stability of the non-PEGylated formulations. The average droplet size for these formulations changed the most for the placebo fish oil NEs (PF). The only statistically significant change (*p* < 0.01) in the PDI could be observed in the same formulation (Table 4). The changes in ZP were significant for all of the non-PEGylated formulations (*p* < 0.01), except for the CF, as were the decrease in pH values (*p* < 0.01) and increase in conductivity (*p* < 0.001) (Table 4). However, the addition of curcumin seemed to cause a smaller decrease in the pH values compared to the placebo formulations. The decrease in the pH value and the increase in the conductivity and droplet size were pronounced the most in PF.

However, the oil phase selection had also a significant effect on the stability of the PEGylated NEs. The use of either PEG2000-DSPE or PEG5000-DPPE led to statistically significant changes in Z-ave, PDI, and ZP for some of the fish oil NEs (Table 6 and Table 8), most noticeably for the formulations with 0.1% of PEG2000-DSPE, both placebo F21 and curcumin-loaded CF21, where all three parameters had a significant change, while those changes were less pronounced in soybean oil formulations (Table 5 and Table 7). The increased stability of soybean oil formulations was further confirmed through pH and conductivity measurements, where a statistically significant difference between values measured initially and after two years of storage was observed in all of the PEGylated formulations (Table 5, Table 6, Table 7 and Table 8), but the changes were less pronounced compared to the fish oil formulations. The decrease in the pH values and conductivity increase in the PEGylated fish oil NEs was more prominent in the placebo formulations than in the curcumin-loaded ones (Table 6 and Table 8). Laser-diffraction measurements preformed on soybean oil samples after two years of storage showed no sign of lager droplet aggregates in all of the soybean oil NEs, with d90 values under 200 nm (Appendix A). With the fish oil formulations, the d90 values were under 200 nm, except for the PF, F53, and F56 NEs (Appendix A). Polarization-microscopy images of the samples taken after two years of storage reveal some bigger droplets, especially in the non-PEGylated samples—CS and CF (Appendix A), with no signs of curcumin crystals in any of the formulations. The content of curcumin changes for some of the soybean oil formulations during storage, namely for CS, CS26, and CS53 (Appendix A) as well as for the CF51 (Appendix A).

Antioxidant assays showed that the use of fish oil instead of soybean oil did not improve the antioxidant potential of the investigated formulations. A DPPH assay demonstrated higher inhibition percentages for curcumin-loaded nanoemulsions compared to the pure curcumin at every investigated concentration (Appendix A). After two years of storage, the antioxidant potential of nanoemulsions remained relatively unchanged (Appendix A), as evidenced by the IC50 values determined initially and after storage (Table 9). There were no significant differences in antioxidant activity observed between soybean (CS, CS21, and CS51) and fish oil formulations (CF, CF21, and CF51), neither initially nor after storage. There were no statistically significant differences in the FRAP values between pure curcumin and curcumin-loaded NEs, while some formulations had higher FRAP values after two years of storage compared to the initially measured ones (Figure 6).

In vitro release studies found a higher release rate of curcumin from the CF formulation compared to the CS at all analyzed time intervals, at a statistically significant level, *p* < 0.05 (Figure 7a). The release of curcumin from the CS formulation is best-described with zero-order kinetics, while the release from the CF formulation best fit the Korsmeyer–Peppas model (Appendix A).

The higher release of curcumin from the fish oil NEs (CF21, CF23, CF26, CF51, CF53, and CF56) compared to the soybean oil ones (CS21, CS23, CS26, CS51, CS53, and CS56) was also noticeable in the PEGylated formulations (Figure 7b,c). The release of curcumin from the soybean oil NEs with PEG2000-DSPE (CS21, CS23, and CS26 with 0.1%, 0.3%, and 0.6% of the PEGylated phospholipid, respectively) was similar (Figure 7b). However, there were some differences between the fish oil formulations, where it could be observed that the addition of 0.1% of PEG2000-DSPE (CF21) slowed down the release of curcumin more pronouncedly than the formulations with 0.3% or 0.6% (CF23 and CF26, respectively). The models that best represented the release of curcumin from the soybean oil NEs were zero-order kinetics for CS23, and Korsmeyer–Peppas for the CS21 and CS26, and for the fish oil NEs: Korsmeyer–Peppas for the CF21 and zero-order kinetics for CF23 and CF26 (Appendix A). In the case of NEs with PEG5000-DPPE, there were no statistically significant differences either between the soybean oil NEs (CS51, CS53, and CS56 with 0.1%, 0.3%, and 0.6% of the PEG-PL, respectively) or between the fish oil ones (CF51, CF53, and CF56 with 0.1%, 0.3%, and 0.6% of the PEG-PL, respectively) (Figure 7c). The release of curcumin from the soybean oil NEs with PEG5000-DPPE (CS51, CS53, CS56) could best be described with the Korsmeyer–Peppas model, while zero-order kinetics best explained the release from the fish oil ones—CF51, CF53, and CF56 (Appendix A).

Syringeability and injectability present critical performance attributes of any parenteral dosage form. Syringeability signifies the ability of an injectable preparation to transfer from a vial through a hypodermic needle prior an injection, while injectability is defined as the force required to inject the formulation from a syringe-needle system into the tissue [28]. The viscosity of nanoemulsions increased with the addition of PEGylated phospholipids, with the viscosity of non-PEGylated formulations about 20 mPa*s, and up to about 60 mPa*s for the formulations with 0.6% of PEG-5000-DPPE, with both oil phases (Figure 8a,c). These viscosity values translated to the injectability values of about 20 N to about 40 N (Figure 8b,d; Appendix A).

## 4. Discussion

The solubility of curcumin in different oils was determined to see if the use of fish oil instead of soybean oil in NEs would impact the amount of curcumin available for incorporation into NEs. The ratio of the soybean/fish oil and MCT mixtures were chosen based on previous experience [22]. It was concluded that, in combination with MCT, soybean oil could be replaced with fish oil, without making changes in the curcumin loading.

For the oil phase composition, more specifically, the presence of different fatty acids could impact the NE stabilization, due to the impact on the interfacial characteristics [29], and subsequently dictate their physicochemical characteristics and biological performance [30]. Therefore, an important aspect of the present study was to elucidate the structure of the formulated NEs using ERP spectroscopy. Three different amphiphilic spin-probes, 5-, 12-, and 16-DSA, were used to determine how the dynamics of the surfactants’ monolayer was affected by the presence of curcumin or the use of different oil phases. The used spin-probes are spin-labeled fatty acid analogs consisting of stearic acid and an N–O• moiety attached to the C-5, C-12, and C-16 positions of the hydrocarbon chain, respectively. Depending on the position of the paramagnetic moiety to the hydrocarbon chain, the spin-probe provides information about different depths of the surfactants’ monolayer. Specifically, the paramagnetic moiety of 5-DSA is localized closer to the polar head groups of the surfactants, while in the case of 16-DSA, it is localized closer to the oil core [31,32]. The decrease in τR, moving from 5-DSA to 12-DSA and 16-DSA, indicated that the 5-DSA spin probe was located in an area where the surfactants were organized in a tighter way, closer to the droplet surface and the aqueous phase and contrary to a more flexible environment moving closer to the oil phase [33], regardless of the oil type used or the presence of curcumin. The EPR analysis indicates that the type of oil used to formulate the NEs plays a crucial role not only in the structure but also in the localization of the bioactive compound. Specifically, curcumin appears to affect the surfactants’ dynamics, as confirmed in previous studies [30,33], in both the soybean oil and the fish oil nanoemulsions; however, curcumin causes a different effect depending on the oil used. In the case of soybean oil, CS, the decrease in τR for the 5-DSA spin probe, compared to the placebo formulation, PS (Table 2), indicated a formation of a less rigid stabilizing layer. The addition of curcumin to fish oil NEs (CF), however, resulted in a more rigid stabilizing layer (higher τR for all probes), indicating a particular impact of curcumin on stabilization of the fish oil NEs. In the case of CS, it appears that the addition of curcumin did not significantly impact the rotation of the other two spin probes, indicating the localization of curcumin in the stabilizing layer closer to the droplet surface and the aqueous phase, while for CF, it appeared that curcumin was distributed throughout the stabilizing layer. The increase in the order parameter for 5-DSA in fish oil NEs could be explained by the decrease in droplet size upon the addition of curcumin (Table 4) and the ensuing increase in the curvature of the stabilizing layer [33]. The hyperfine coupling constant (αN) provides information regarding the polarity profile across the surfactant monolayer. In the case of fish oil nanoemulsions, the addition of curcumin results in decreased polarity in the surfactant region closer to the aqueous phase (Table 2). Bearing in mind curcumin’s impact on the NE’s stabilizing layer, it would be interesting to see how additional stabilizers and droplet surface modifiers (PEGylated phospholipids) impact the characteristics and the stability of ensuing formulations.

One of the most important characteristics of any nanoproduct is their size [34,35]. In the case of pharmaceuticals intended for parenteral delivery of drugs, the importance of this property is multifold—it impacts their stability, pharmacokinetics upon administration, and safety [35,36,37]. It is suggested that the optimal droplet size for parenteral nanoemulsions is below 500 nm [38,39]. On the other hand, droplet-size distribution is a critical parameter that provides information on the homogeneity of the formulation and its quality [40], and for parenteral preparations these values should be kept below 0.25 [41]. Many composition and preparation factors impact NE droplet size and droplet-size distribution, such as the oil phase selection and ratio and high pressure homogenization cycles and pressure [37,42,43]. In this research, we employed the design of experiments strategy in order to gain better understanding regarding the combined impact of different formulation factors (PEG-PL type and concentration, oil phase selection, and presence of curcumin) on NEs’ Z-ave and PDI. This approach allowed for a better estimation of how these factors impact the critical quality attributes of the PEGylated nanoemulsions (Z-ave and PDI), compared to varying just one factor at the time, because it could detect interactions between different factors that could otherwise go unnoticed.

The experimental design study stressed out the importance of considering simultaneous impact of factors on CQAs of NEs [44]. In the present research, the most prominent interactions were between the PEG-PL type and concentration (AB) and between the concentration of the PEG-PLs and the presence of curcumin (BD), which impact both Z-ave and PDI (Figure 2 and Figure 3). This study showed that the presence of curcumin had an impact on droplet-size reduction, which reflected the EPR structural analysis findings regarding the presence of curcumin in the NE stabilizing layer and helped affirmation of its potentially stabilizing properties [45]. Additionally, it confirmed the oil phase impact on Z-ave, evident in the EPR structural analysis. This could be attributed to the fatty oil composition of the oils, specifically to oleic acid, which is present in a higher concentration in soybean oil and could be used as an additional stabilizer in NEs [27,46], accounting for the smaller droplets in soybean oil NEs. Although Z-ave was predominantly influenced by individual factors, the quality of nanoemulsions was impacted more by the interactions between the investigated factors, specifically implying that in order to obtain NEs with the smallest PDI, careful considerations should be applied regarding the selection of the PEGylated phospholipids, their concentration, and the adequate oil phase.

The initial values of selected physicochemical parameters (Table 4, Table 5, Table 6, Table 7 and Table 8) indicated that all formulations were suitable for parenteral administration [39,40,41,47]. The stability studies were performed in order to assess the long-term impact of the experimental design input parameters on the NEs’ characteristics. The addition of co-emulsifiers can have beneficial or disadvantageous effects on droplets and the release of the incorporated active. The additional stabilizers could lead to tighter packing in the interfacial layer, leading to slower release of the active from the droplets and better stability, or they could disturb the droplet integrity [29]. The biggest increase in droplet size for the PF NE, compared to the other non-PEGylated formulations (CF, PS, and CS) (Table 4), indicated that the addition of curcumin to the fish oil NEs and its location in the interfacial layer, as confirmed by the EPR study, led to increased NE stability. The change of ZP value in PF during storage further proved the stabilizing effect of curcumin in the interface layer of fish oil NEs, for this change was not noticeable in the curcumin-loaded NE—CF. The increase in absolute zeta potential values (Table 4) could be explained by the hydrolysis of triglycerides included in lecithin, namely the formation of free fatty acids [29]. These changes promoted further degradation of the stabilizing layer and destabilization of the NEs. As the products of lecithin hydrolysis are more water-soluble, this accounted for the decrease in pH and the increase in conductivity values (Table 4). The decrease in pH and increase in conductivity values in the curcumin-loaded formulations (CS an CF) was less pronounced compared to the placebo counterparts (PS and PF) (Table 4), which further accentuated the claims of curcumin’s stabilizing properties. Overall, the changes in the stability parameters identified in PF suggested that the addition of curcumin and its localization in the stabilizing layer had a particular stabilizing effect in the fish oil NEs, corresponding to the EPR findings.

The addition of the PEGylated phospholipids, regardless of the oil type or the presence of curcumin, led to increased stability compared to the non-PEGylated formulations (Table 4, Table 5, Table 6, Table 7 and Table 8). This was reflected in the droplet-size measurements, where only a slight increase in Z-ave (less than 10 nm) could be observed after two years of storage. The long-term stabilizing properties of the PEGylated phospholipids were especially significant in the fish oil NEs (both placebo F21, F23, F26, F51, F53, and F56 and curcumin-loaded CF21, CF23, CF26, CF51, CF53, and CF56), as evidenced by the pH and conductivity changes, specifically when compared to the non-PEGylated formulations. The impact of curcumin on the stabilization of fish oil NEs was further confirmed with the LD measurements, with bigger droplets observed only in the placebo formulations—PF, F53, and F56 (Appendix A). The absence of bigger droplets in soybean oil formulations (both placebo S21, S23, S26, S51, S53, and S56 and curcumin-loaded CS21, CS23, CS26, CS51, CS53, and CS56) after two years (Appendix A) further underlines the significance of the oil phase selection. The addition of the PEGylated phospholipids did not impact the curcumin’s encapsulation, which was >99% for the PEGylated NEs.

The higher inhibition percentages for curcumin-loaded NEs in the DPPH antioxidant assay, as opposed to pure curcumin (Appendix A), could be attributed to the NEs composition (Table 1), more specifically to the BHT, which was added to the oil phase to prevent the oxidation of lecithin. The preserved antioxidant activity of curcumin in the NEs correlates with the curcumin content stability during storage (Appendix A), further proving the long-term stability of the formulations. Interestingly, fish oil itself did not provide additional antioxidant boost to the NEs compared to soybean oil, as could possibly be expected given its fatty acid composition, particularly docosahexaenoic acid (DHA) and eicosapentaenoic acid (EPA) [48,49]. DPPH assay entails the destruction of the droplet structure, given the organic solvent environment, so the contribution of the NE system itself to the antioxidant activity could not be adequately assessed by only using this method. On the other hand, the FRAP test takes place in an aqueous medium where the droplet structure is preserved. The lack of difference in the antioxidant activity between pure curcumin and curcumin-loaded NEs could probably be accounted for by the location of curcumin in the stabilizing layer of the droplets, and, therefore, its availability to readily participate in the reduction of ferric tripyridyltriazine complex to the ferrous tripyridyltriazine. Interestingly, most formulations showed increased FRAP values after storage (CS, CS21, CF, CF21, and CF51) (Figure 6). This could probably be explained by the fact that they were measured against corresponding placebos, stored for the same time period, as blanks. Therefore, seemingly increased FRAP values after storage could probably be attributed to the decreased antioxidant potential of the placebos (as evidenced by their physicochemical characteristics, Table 4, Table 5, Table 6, Table 7 and Table 8), which gave the blanks lower absorbance values compared to the initially measured ones and the curcumin-loaded NEs an apparent absorbance boost, resulting in greater FRAP values compared to the initial ones. Similarly to the DPPH assay, the oil phase selection did not seem to impact the antioxidant stability of the formulations, given that there were no significant differences in FRAP values between the soybean oil and fish oil formulations. FRAP assay also confirmed the maintained antioxidant characteristics of the formulations during storage.

A higher release of curcumin was noticed from the fish oil NEs compared to the soybean oil ones. This could be explained by the fact that curcumin has a bigger stabilizing impact on the fish oil NEs, as evidenced by the EPR study. Therefore, when fish oil NEs are found in the release medium, the exit of curcumin from the formulations has a more catastrophic impact on fish oil NEs, causing higher release rates from these formulations. The release of curcumin from CS and CF formulations was best-described by the zero-order kinetics and the Korsmeyer–Peppas model, respectively, given that this model showed the highest R^2^_adj_ value and the lowest AIC (Akaike Information Criterion) value [50]. The release exponent value N in the Korsmeyer–Peppas model of the CF formulation (Appendix A) indicated anomalous transport, where the mechanism of drug release is governed by swelling and diffusion [51]. It should be noted that the release of curcumin from these formulations could also be described with some other models that had R^2^_adj_ values > 0.99, indicating a complex release for both formulations (Appendix A). The addition of different concentrations of PEG2000-DSPE to the soybean oil formulations did not seem to have a major impact on the drug release (Figure 7b). This could potentially be explained by the fact that optimal surface protection of the NE droplets was achieved with lowest concentration of the PEGylated phospholipid, so a further increase in its concentration did not have an impact on drug release. On the other hand, it appeared that 0.1% of the PEG2000-DSPE slowed down the release of curcumin the most for the fish oil NEs, compared to the other two concentrations of the same PEGylated PL. It could be hypothesized that further addition of the PEGylated phospholipid to this formulation actually increased the release curcumin, by decreasing the rigidity of the stabilizing layer. Similarly to the non-PEGylated NEs, the dissolution data modeling for the formulations with PEG2000-DSPE indicated a complex drug releasing mechanisms, where in some cases several models could be used to describe the drug release from the formulations (Appendix A). The lack of statistically significant differences in the curcumin released from the formulations containing PEG5000-DPPE, either with soybean or fish oil, possibly indicated that the optimal surface coverage was achieved using the lowest concentrations of the PEGylated phospholipid, 0.1%, and that the further addition potentially only decreased the integrity of the stabilizing layer. The release of curcumin from the CS51, CS53, and CS56 NEs best fit the Korsmeyer–Peppas model, with the release exponent being higher than 1 in all cases, indicating Super Case-II transport (Appendix A). In this case, the velocity of the solvent diffusion is higher than the polymer relaxation process, causing an acceleration of solvent penetration into the droplet [51]. The release of curcumin from the CF51, CF53, and CF56 NEs follows the zero-order kinetics, where the same amount of drug is released by unit of time, which is considered ideal to achieve prolonged drug release [50].

The rheological characteristics of nanoemulsions determine their behavior during administration but could also impact the safety upon administration and limit their use as dosage forms [52]. There are no specific requirements by the USP or Ph. Eur. Pharmacopoeias, in regards to viscosity of the intravenous preparations. Given that parenteral administration route is one of the most frequently used [53,54] and that various polyethylene glycols (PEGs) are customarily used as rheology modifiers in parenteral suspensions, it was important to determine how the addition of PEGylated phospholipids impacted the viscosity and subsequent physical performance of the developed systems. To the best of our knowledge, there are no methods for injectability assessment for intravenous preparations described in the literature. For intravenous administration of test formulations, in our laboratory setting we use a syringe pump (Stoelting Co., Wood Dale, IL, USA) that delivers the preparation through 10 mL syringe and a 25 G scalp vein infusion set (Romed, Wilnis, The Netherlands). For the injectability testing, the probe of the texture analyzer was set to move downward at 1 mm/s speed, and the formulations were extruded into the blood-mimicking glycerol solution (36.6%, *v*/*v* in water; [55]), with the viscosity value of 4726 mPa*s, circulating at about 4 mL/min speed through a silicone tube (inner diameter of 2 mm), representing blood circulating through a blood vessel. This setting was designed not only to test the force needed to administer the formulations into blood but also to detect possible clogging as a result of the formulations’ viscosity. The addition of the PEGylated phospholipids caused an increase in viscosity for NEs, regardless of the oil phase selection (Figure 8a,c). The same trend could be observed in the injectability testing (Figure 8b,d), where increased force was required to extrude formulations with increased viscosity. Even though there are no requirements limiting the injection force of the formulation, some previous work regarding the subcutaneous administration [54] suggested a maximum injection force of 40 N and recommended targeting no more than 20 N. It could be observed from Appendix A that the highest force achieved before the content was extruded (F_max_) was below 40 N for all of the formulations and about 20 N for the formulations containing 0.1% of the PEG2000-DSPE (CS21 and CF21) as well as for the non-PEGylated ones (CS and CF). Additionally, none of the analyzed NEs showed any sign of clogging the silicone tube, indicating suitability for intravenous application. Based on these observations, it could be suggested that the viscosity values of about 60 mPa*s could be used in prospective animal studies for intravenous application, with the optimal viscosity below 20 mPa*s.

## 5. Conclusions

With the presenting study, we showed that the oil phase selection played a vital role for the characteristics of the stabilizing layer, since a more rigid stabilizing structure was achieved using soybean oil compared to fish oil. The addition of curcumin impacted the stabilizing layer, having different effects depending on the type of oil—promoting rigidity for the fish oil formulations, while reducing it for the soybean oil ones. In addition, the smallest droplets with the narrowest droplet-size distribution were achieved with curcumin-loaded soybean oil formulations, underlining the role of curcumin in NE stabilization. Stability testing performed after two years of storage at room temperature accentuated the contribution of the PEGylated phospholipids to the NE’s stability, with a better stability profile, in terms of less significant pH decrease and conductivity increase, observed for the PEGylated NEs. DPPH and FRAP assays confirmed the maintained antioxidant activity of the formulations during storage. In vitro release testing demonstrated increased drug release from the fish oil NEs compared to the soybean oil ones, while dissolution data modeling indicated complex release mechanisms from the formulations. However, oil phase selection did not have a significant effect on injectability, thus, all of the formulations could be good candidates for prospective in vivo animal trials, with a clear advantage given to the formulations containing 0.1% of either PEG2000-DSPE or PEG5000-DPPE. Overall, the study proved the impact of proper oil phase selection on the structure, physicochemical characteristics, long-term stability, and performance of PEGylated NEs.

## Figures and Tables

**Figure 1 pharmaceutics-14-01666-f001:**
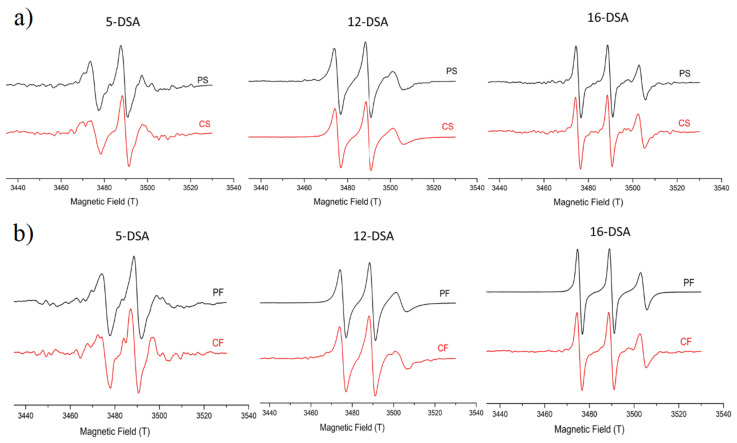
EPR spectra of the three spin-probes in (**a**) soybean and (**b**) fish oil NEs in the presence (red line) and absence (black line) of curcumin.

**Figure 2 pharmaceutics-14-01666-f002:**
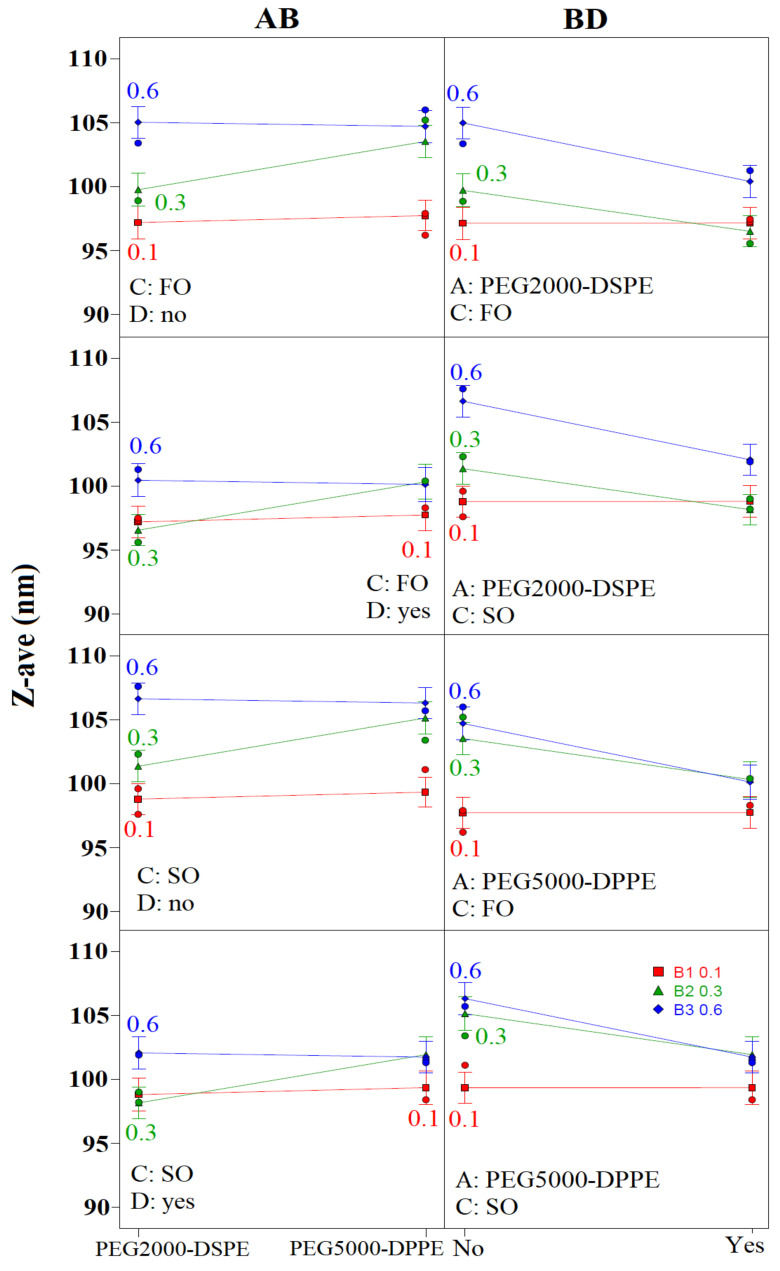
Interaction plot for the mean droplet size (Z-ave). FO: fish oil; SO: soybean oil. AB—PEG-PL type/PEG-PL concentration; BD—PEG-PL concentration/presence of curcumin.

**Figure 3 pharmaceutics-14-01666-f003:**
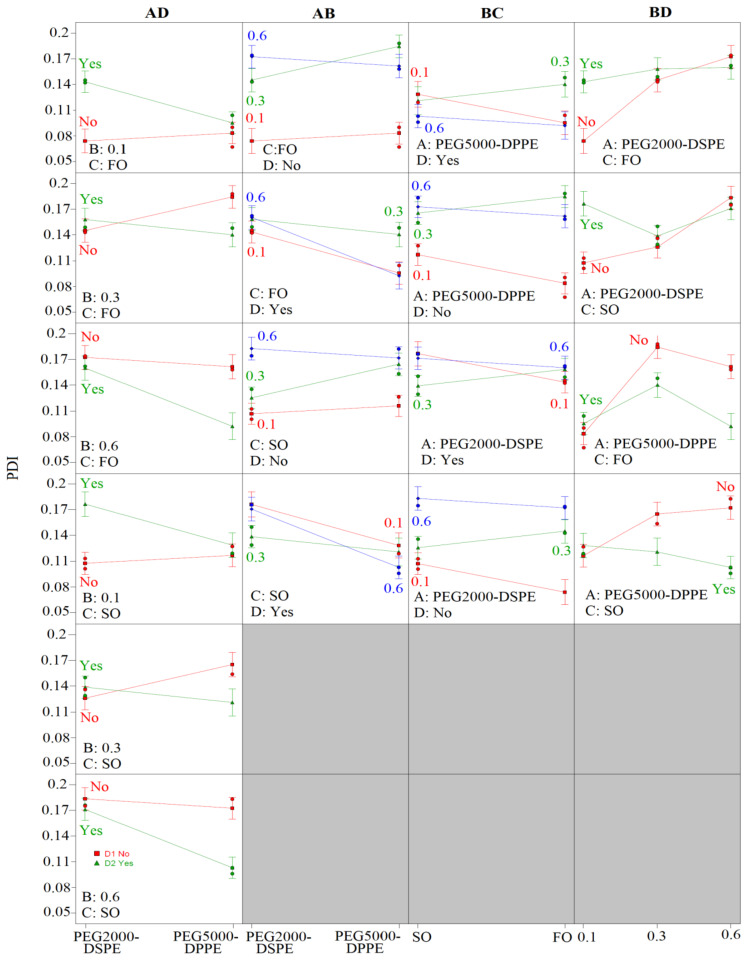
Interaction plot for the polydispersity index (PDI). FO: fish oil; SO: soybean oil. AD—PEG-PL type/presence of curcumin; AB—PEG-PL type/PEG-PL concentration; BC—PEG-PL concentration/oil type; BD—PEG-PL concentration/presence of curcumin.

**Figure 4 pharmaceutics-14-01666-f004:**
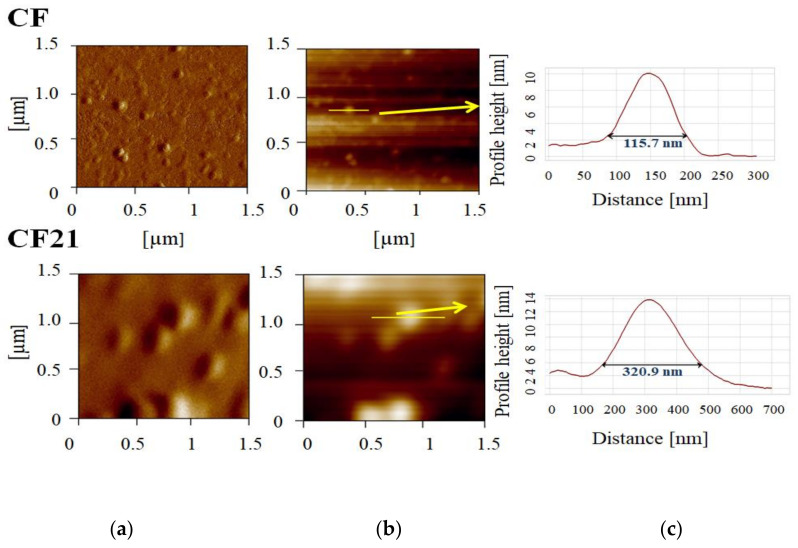
AFM images of curcumin-loaded nanoemulsions with fish oil, non-PEGylated (CF), and PEGylated containing 0.1% of PEG2000-DSPE (CF21): (**a**) error signal of 1.5 × 1.5 µm^2^ area of the sample; (**b**) 2D topography; (**c**) height profiles of selected nanoemulsion droplets.

**Figure 5 pharmaceutics-14-01666-f005:**
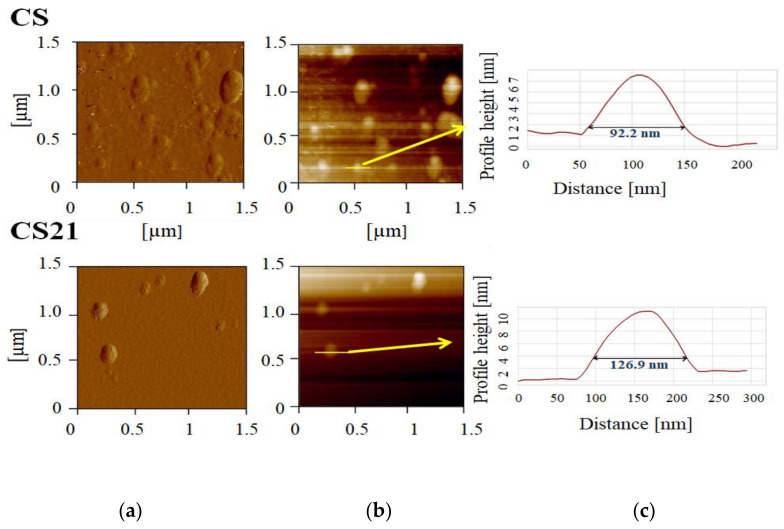
AFM images of curcumin-loaded nanoemulsions with soybean oil, non-PEGylated (CS), and PEGylated containing 0.1% of PEG2000-DSPE (CS21): (**a**) error signal of 1.5 × 1.5 µm^2^ area of the sample; (**b**) 2D topography; (**c**) height profiles of selected nanoemulsion droplets.

**Figure 6 pharmaceutics-14-01666-f006:**
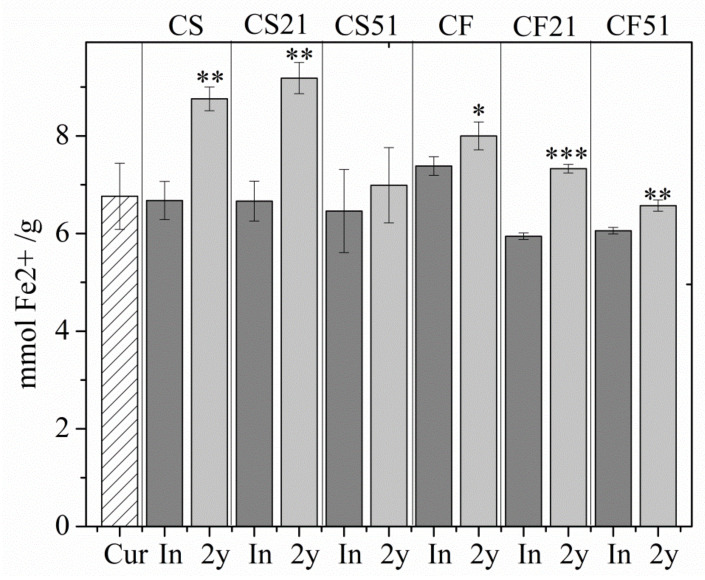
FRAP assay values for formulations after preparation (In) and after two years of storage (2 y). Values are shown as means ± sd (*n* = 3); *, ** and ***, *p* < 0.05; *p* < 0.01 and *p* < 0.001 compared to the initially measured values.

**Figure 7 pharmaceutics-14-01666-f007:**
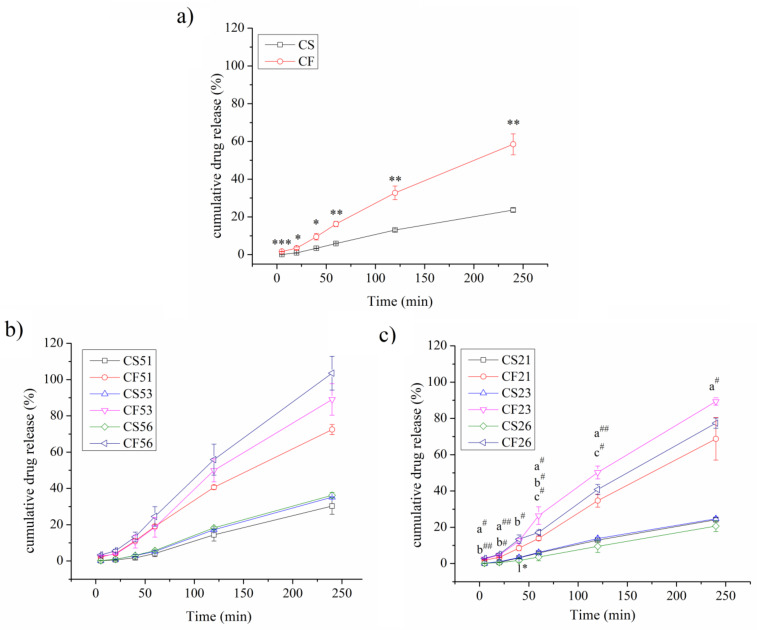
Cumulative release of curcumin from (**a**) non-PEGylated formulations with soybean oil (CS) and fish oil (CF); (**b**) NEs with PEG2000-DSPE containing soybean oil (CS21, CS23, and CS26) or fish oil (CF21, CF23, and CF26); (**c**) NEs with PEG5000-DPPE containing soybean oil (CS51, CS53, and CS56) or fish oil (CF51, CF53, and CF56); values are shown as means ± sd (*n* = 3); *, **, and ***, *p* < 0.05, *p* < 0.01, and *p* < 0.001 compared to CS; a^#^
*p* < 0.05, a^##^
*p* < 0.01, CF21 vs. CF23; b^#^, b^##^, *p* < 0.05 and *p* < 0.01 CF21 vs. CF26; c^#^
*p* < 0.05, CF23 vs. CF26; 1* *p* < 0.05, CS23 vs. CS26.

**Figure 8 pharmaceutics-14-01666-f008:**
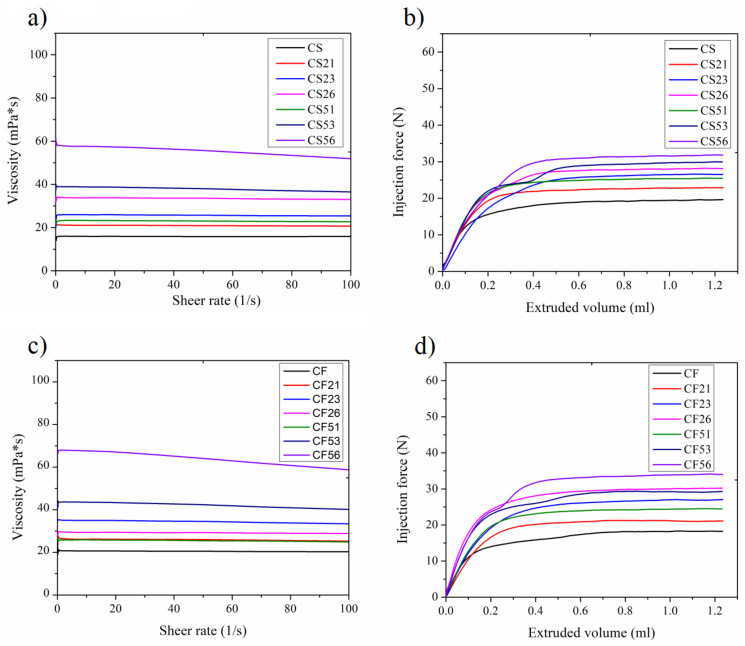
(**a**) Viscosity and (**b**) injectability for soybean oil nanoemulsions (CS21, CS23, and CS26 with 0.1%, 0.3%, and 0.6% of PEG2000-DSPE, respectively; CS51, CS53, and CS56 with 0.1%, 0.3%, and 0.6% of PEG5000-DPPE, respectively); (**c**) viscosity and (**d**) injectability for fish oil nanoemulsions (CF21, CF23, and CF26 with 0.1%, 0.3%, and 0.6% of PEG2000-DSPE, respectively; CF51, CF53, and CF56 with 0.1%, 0.3%, and 0.6% of PEG5000-DPPE, respectively).

**Table 1 pharmaceutics-14-01666-t001:** Composition of placebo and curcumin-loaded nanoemulsions.

Nanoemulsions														
Ingredients (%, *w*/*w*)	PS/ PF	S21/ F21	S23/ F23	S26/ F26	S51/ F51	S53/ F53	S56/ F56	CS/ CF	CS21/ CF21	CS23/ CF23	CS26/ CF26	CS51/ CF51	CS53/ CF53	CS56/ CF56
**Oil phase**														
**Curcumin**	-	-	-	-	-	-	-	0.075	0.075	0.075	0.075	0.075	0.075	0.075
**SO/FO**	4	4	4	4	4	4	4	4	4	4	4	4	4	4
**MCT**	16	16	16	16	16	16	16	16	16	16	16	16	16	16
**LS**	2.3	2.3	2.3	2.3	2.3	2.3	2.3	2.3	2.3	2.3	2.3	2.3	2.3	2.3
**BA**	3	3	3	0	3	3	3	3	3	3	3	3	3	3
**BHT**	0.05	0.05	0.05	0.05	0.05	0.05	0.05	0.05	0.05	0.05	0.05	0.05	0.05	0.05
**PEG2000-DSPE**	-	0.1	0.3	0.6	-	-	-	-	0.1	0.3	0.6	-	-	-
**Aqueous phase**														
**P80**	2.3	2.3	2.3	2.3	2.3	2.3	2.3	2.3	2.3	2.3	2.3	2.3	2.3	2.3
**Glycerol**	0.8	0.8	0.8	0.8	0.8	0.8	0.8	0.8	0.8	0.8	0.8	0.8	0.8	0.8
**NaO**	0.03	0.03	0.03	0.03	0.03	0.03	0.03	0.03	0.03	0.03	0.03	0.03	0.03	0.03
**PEG5000-DPPE**	-	-	-	-	0.1	0.3	0.6	-	-	-	-	0.1	0.3	0.6
**Water to**	100	100	100	100	100	100	100	100	100	100	100	100	100	100

SO: soybean oil; FO: fish oil; MCT: medium chain triglycerides; BA: benzyl alcohol; BHT: butylated hydroxytoluene; LS: soybean lecithin; P80: polysorbate 80; NaO: sodium oleate. Soybean oil-loaded nanoemulsions: **non-PEGylated:** PS—placebo; CS—curcumin-loaded; **with PEG2000-DSPE in 0.1%, 0.3%, and 0.6% concentrations:** placebo—S21, S23, and S26; curcumin-loaded—CS21, CS23, and CS26; **with PEG5000-DPPE in 0.1%, 0.3%, and 0.6% concentrations:** placebo—S51, S53, and S56; curcumin-loaded—CS51, CS53, and CS56. F: fish oil-loaded nanoemulsions: **non-PEGylated:** PF—placebo; CF—curcumin-loaded, **with PEG2000-DSPE in 0.1%, 0.3%, and 0.6% concentrations:** placebo—F21, F23, and F26; curcumin-loaded—CF21, CF23, and CF26; **with PEG5000-DPPE in 0.1%, 0.3%, and 0.6% concentrations:** placebo—F51, F53, and F56; curcumin-loaded—CF51, CF53, and CF56.

**Table 2 pharmaceutics-14-01666-t002:** Rotational correlation time (τR), order parameter (S), and isotropic hyperfine coupling constant (αN) of 5-DSA, 12-DSA, and 16-DSA in empty and loaded nanoemulsions with soybean (PS, CS) and fish oil (PF, CF).

Spin-Probes	EPR Values	Formulations			
		PS	CS	PF	CF
5-DSA	τR (ns)	2.18 ± 0.60	1.66 ± 0.61	1.19 ± 0.10	2.96 ± 0.81
	S	0.22 ± 0.04	0.29 ± 0.11	0.17 ± 0.04	0.26 ± 0.03
	αN (×10^−4^ T)	12.79 ± 0.37	13.14 ± 0.80	14.17 ± 0.38	12.79 ± 0.52
12-DSA	τR (ns)	1.95 ± 0.07	1.81 ± 0.06	1.63 ± 0.13	2.27 ± 0.19
	S	0.11 ± 0.001	0.11 ± 0.01	0.10 ± 0.01	0.13 ± 0.001
	αN (×10^−4^ T)	14.29 ± 0.55	14.08 ± 0.02	14.33 ± 0.10	14.33 ± 0.11
16-DSA	τR (ns)	0.63 ± 0.02	0.61 ± 0.03	0.55 ± 0.07	0.72 ± 0.07
	S	0.05 ± 0.001	0.05 ± 0.001	0.04 ± 0.001	0.04 ± 0.001
	αN (×10^−4^ T)	14.77 ± 0.02	14.64 ± 0.01	14.75 ± 0.05	14.67 ± 0.06

**Table 3 pharmaceutics-14-01666-t003:** D-optimal factorial design matrix.

Run	Variables							Responses	
	A: PL Type		B: PL Concentration	C: Oil Type		D: Model Drug (Curcumin)		Z-ave (nm)	PDI
	AL	CL	AL	AL	CL	AL	CL		
1 *	PEG200−0-DSPE	−1	0.1%	MCT-SO	−1	No	−1	97.58 ± 1.82	0.11 ± 0.01
2 *				MCT-FO	+1	Yes	+1	97.33 ± 0.17	0.15 ± 0.00
3			0.3%	MCT-SO	−1	No	−1	102.30 ± 2.46	0.14 ± 0.03
4 *						Yes	+1	99.03 ± 0.87	0.15 ± 0.02
5				MCT-FO	+1	No	−1	98.86 ± 1.34	0.14 ± 0.01
6						Yes	+1	95.56 ± 0.58	0.15 ± 0.02
7			0.6%	MCT-SO	−1	No	−1	107.63 ± 1.82	0.18 ± 0.00
8						Yes	+1	101.87 ± 1.52	0.18 ± 0.03
9				MCT-FO	+1	No	−1	103.40 ± 1.23	0.17 ± 0.03
10						Yes	+1	101.33 ± 0.35	0.16 ± 0.02
11	PEG5000-DPPE	+1	0.1%	MCT-SO	−1	No	−1	101.13 ± 1.75	0.13 ± 0.01
12						Yes	+1	98.41 ±0.64	0.12 ± 0.02
13 *				MCT-FO	+1	No	−1	96.21 ± 1.51	0.07 ± 0.02
14						Yes	+1	98.28 ± 1.14	0.10 ± 0.04
15			0.3%	MCT-SO	−1	No	−1	103.37 ± 1.95	0.15 ± 0.02
16				MCT-FO	+1	No	−1	105.17 ± 0.31	0.19 ± 0.02
17						Yes	+1	100.43 ± 0.29	0.15 ± 0.02
18			0.6%	MCT-SO	−1	No	−1	105.73 ± 1.78	0.18 ± 0.02
19 *						Yes	+1	101.47 ± 0.29	0.10 ± 0.01
20				MCT-FO		No	−1	105.97 ± 2.35	0.16 ± 0.01

* Replicated runs; AL: actual level; CL: coded level; MCT-medium chain triglycerides; SO—soybean oil; FO—fish oil.

**Table 4 pharmaceutics-14-01666-t004:** Physicochemical parameters for non-PEGylated nanoemulsions, with soybean oil (PS—placebo and CS—curcumin-loaded) and fish oil (PF—placebo and CF—curcumin-loaded) measured initially (In) and after two years of storage at room temperature (2 y).

		PS	CS	PF	CF
Z-ave (nm)	In	98.54 ± 2.32	95.57 ± 0.32	99.57 ± 2.42	94.35 ± 1.68
	2 y	130.90 ± 1.65 ***	131.10 ± 1.65 ***	283.40 ± 1.88 ***	126.20 ± 1.99 ***
PDI	In	0.13 ± 0.01	0.11 ± 0.03	0.13 ± 0.01	0.09 ± 0.00
	2 y	0.11 ± 0.02	0.10 ± 0.02	0.06 ± 0.02 **	0.09 ± 0.01
ZP (mV)	In	−43.00 ± 1.25	−31.80 ± 1.30	−42.20 ± 0.76	−44.10 ± 0.57
	2 y	−53.50 ± 1.25 **	−46.50 ± 1.37 ***	−51.50 ± 0.50 ***	−44.60 ± 0.35
pH	In	7.31 ± 0.01	6.95 ± 0.05	7.41 ±0.01	7.37 ± 0.03
	2 y	4.95 ± 0.01 ***	5.52 ± 0.01 ***	3.35 ± 0.01 ***	4.06 ± 0.01 ***
Conductivity (µS/cm)	In	173.93 ± 0.67	174.17 ± 2.40	166.07 ± 1.10	181.07 ± 0.61
	2 y	205.67 ± 2.08 ***	182.67 ± 1.00 **	458.67 ± 1.53 ***	196.20 ± 0.61 ***

Values are shown as means ± sd (*n* = 3); ** and ***, *p* < 0.01 and *p* < 0.001, compared to the initially measured values.

**Table 5 pharmaceutics-14-01666-t005:** Physicochemical parameters for soybean oil nanoemulsions containing PEG2000-DSPE measured initially (In) and after two years of storage at room temperature (2 y).

		S21	CS21	S23	CS23	S26	CS26
Z-ave (nm)	In	97.58 ± 1.82	95.86 ± 1.17	102.30 ± 2.46	99.03 ± 0.87	107.60 ± 1.82	101.90 ± 1.52
	2 y	101.10 ± 2.69	107.60 ± 1.88 **	102.40 ± 0.67	99.74 ± 1.15	105.90 ± 1.55	102.70 ±0.95
PDI	In	0.11 ± 0.01	0.12 ± 0.02	0.14 ± 0.03	0.15 ± 0.02	0.18 ± 0.00	0.18 ± 0.03
	2 y	0.06 ± 0.01 *	0.13 ± 0.02	0.13 ± 0.02	0.12 ± 0.02	0.17 ± 0.02	0.17 ± 0.01
ZP (mV)	In	−41.00 ± 0.99	−39.10 ± 0.17	−44.00 ± 1.07	−37.00 ± 1.39	−40.60 ± 0.60	−42.20 ± 2.50
	2 y	−49.30 ± 3.39 *	−44.20 ± 2.82	−44.90 ± 0.21	−42.00 ± 0.59 **	−49.80 ± 2.25 *	−43.00 ± 1.31
pH	In	7.34 ±0.01	7.08 ± 0.01	7.42 ± 0.03	6.78 ± 0.03	7.34 ± 0.02	7.27 ± 0.02
	2 y	5.44 ± 0.01 ***	5.44 ± 0.00 ***	5.58 ± 0.01 ***	5.54 ± 0.01 ***	5.31 ± 0.01 ***	5.62 ± 0.03 ***
Conductivity (µS/cm)	In	176.17 ± 1.55	186.33 ± 0.74	198.67 ± 1.53	223.33 ± 1.53	220.00 ± 1.00	241.67 ± 2.52
	2 y	212.67 ± 2.08 ***	204.67 ± 3.51 **	213.00 ± 1.73 ***	228.67 ± 1.53*	237.33 ± 1.53 ***	258.33 ± 2.31 **

S21—placebo and CS21—curcumin loaded with 0.1%; S23—placebo and CS23—curcumin loaded with 0.3%; S26—placebo and CS26—curcumin loaded with 0.6% of the PEG-PL. Values are shown as means ± sd (*n* = 3); *, **, and ***, *p* < 0.05; *p* < 0.01, and *p* < 0.001 compared to the initially measured values.

**Table 6 pharmaceutics-14-01666-t006:** Physicochemical parameters for fish oil nanoemulsions containing PEG2000-DSPE measured initially (In) and after two years of storage at room temperature (2 y).

		F21	CF21	F23	CF23	F26	CF26
Z-ave (nm)	In	98.23 ± 2.02	97.33 ± 0.17	98.86 ± 1.34	95.56 ± 0.58	103.40 ± 1.23	103.30 ± 0.35
	2 y	102.10 ± 1.22 *	104.20 ± 1.36 **	103.10 ± 1.14 *	99.35 ± 1.31 *	109.10 ± 1.80 *	105.90 ± 1.06 **
PDI	In	0.15 ± 0.01	0.15 ± 0.00	0.14 ± 0.01	0.15 ± 0.02	0.17 ± 0.03	0.16 ± 0.02
	2 y	0.08 ± 0.01 **	0.09 ± 0.01 ***	0.11 ± 0.02	0.10 ± 0.01 *	0.13 ± 0.03	0.13 ± 0.03
ZP (mV)	In	−40.20 ± 0.47	−42.40 ± 1.05	−37.60 ± 0.55	−47.30 ± 2.72	−47.70 ± 2.58	−42.40 ± 1.34
	2 y	−48.10 ± 0.23 ***	−45.30 ± 1.46 *	−55.90 ± 1.40 ***	−45.40 ± 0.92	−53.70 ± 0.76 *	−47.90 ± 1.97 *
pH	In	7.46 ± 0.03	7.36 ± 0.02	7.43 ± 0.01	7.31 ± 0.01	7.26 ± 0.05	7.34 ± 0.03
	2 y	3.39 ± 0.01 ***	3.97 ± 0.00 ***	3.30 ± 0.00 ***	3.85 ± 0.02 ***	3.28 ± 0.01 ***	3.64 ± 0.02 ***
Conductivity (µS/cm)	In	177.07 ± 1.11	187.40 ± 1.00	212.00 ± 0.00	203.00 ± 2.00	229.67 ± 2.52	235.33 ± 4.04
	2 y	280.00 ± 2.65 ***	217.33 ± 2.08 ***	318.67 ± 0.58 ***	264.67 ± 2.52 ***	357.00 ± 1.00 ***	317.00 ± 2.65 ***

F21—placebo and CF21—curcumin loaded with 0.1%; F23—placebo and CF23—curcumin loaded with 0.3%; F26—placebo and CF26—curcumin loaded with 0.6% of the PEG-PL. Values are shown as means ± sd (*n* = 3); *, **, and ***, *p* < 0.05, *p* < 0.01, and *p* < 0.001 compared to the initially measured values.

**Table 7 pharmaceutics-14-01666-t007:** Physicochemical parameters for soybean oil nanoemulsions containing PEG5000-DPPE measured initially (In) and after two years of storage at room temperature (2 y).

		S51	CS51	S53	CS53	S56	CS56
Z-ave (nm)	In	101.10 ± 1.75	98.41 ± 0.64	103.40 ± 1.95	97.89 ± 0.61	105.70 ± 1.78	101.50 ± 0.29
	2 y	110.60 ± 1.66 **	109.60 ± 1.28 ***	103.10 ± 0.38	98.34 ± 1.25	99.93 ± 1.11 **	105.70 ± 0.81 **
PDI	In	0.13 ± 0.01	0.12 ± 0.02	0.15 ± 0.02	0.12 ± 0.03	0.18 ± 0.02	0.10 ± 0.01
	2 y	0.13 ± 0.00	0.09 ± 0.02	0.13 ± 0.01	0.10 ± 0.01	0.08 ± 0.03 **	0.16 ± 0.01 **
ZP (mV)	In	−37.30 ± 1.35	−37.50 ± 1.60	−32.50 ± 1.39	−34.20 ± 0.75	−29.40 ± 1.35	−32.40 ± 1.67
	2 y	−46.70 ± 1.16 **	−38.50 ± 0.61	−43.80 ± 0.49 ***	−49.30 ± 0.83 ***	−36.70 ± 1.15 **	−33.80 ± 0.57
pH	In	7.26 ± 0.01	7.14 ± 0.02	7.32 ± 0.02	7.17 ± 0.01	7.43 ± 0.02	7.40 ± 0.01
	2 y	4.70 ± 0.01 ***	5.55 ± 0.01 ***	5.39 ± 0.01 ***	5.60 ± 0.01 ***	5.41 ± 0.01 ***	5.69 ± 0.01 ***
Conductivity (µS/cm)	In	165.37 ± 0.93	176.03 ± 0.75	173.10 ± 1.21	181.80 ± 0.50	179.90 ± 1.93	198.40 ± 1.01
	2 y	190.87 ± 1.07 ***	186.17 ± 0.87 ***	200.10 ± 1.85 ***	211.67 ± 1.15 ***	209.67 ± 3.06 ***	211.33 ± 1.15 ***

S51—placebo and CS51—curcumin loaded with 0.1%; S53—placebo and CS53—curcumin loaded with 0.3%; S26—placebo and CS56—curcumin loaded with 0.6% of the PEG-PL. Values are shown as means ± sd (*n* = 3); ** and ***, *p* < 0.01 and *p* < 0.001 compared to the initially measured values.

**Table 8 pharmaceutics-14-01666-t008:** Physicochemical parameters for fish oil nanoemulsions containing PEG5000-DPPE measured initially (In) and after two years of storage at room temperature (2 y).

		F51	CF51	F53	CF53	F56	CF56
Z-ave (nm)	In	96.21 ± 1.51	98.28 ± 1.14	105.20 ± 0.31	100.40 ± 0.30	106.00 ± 2.35	98.43 ± 0.58
	2 y	107.40 ± 1.25 **	109.60 ± 2.25 **	105.60 ± 2.37	99.91 ± 1.67	106.80 ± 1.01	101.40 ± 1.10 *
PDI	In	0.07 ± 0.02	0.10 ± 0.04	0.19 ± 0.02	0.15 ± 0.02	0.16 ± 0.01	0.09 ± 0.02
	2 y	0.09 ± 0.03	0.08 ± 0.02	0.12 ± 0.03 *	0.11 ± 0.01 *	0.12 ± 0.02 *	0.11 ± 0.02
ZP (mV)	In	−39.00 ± 1.03	−38.30 ± 1.79	−35.90 ± 2.02	−38.90 ± 4.62	−32.60 ± 1.15	−35.20 ± 0.80
	2 y	−54.60 ± 0.91 ***	−44.30 ± 0.46 **	−48.70 ± 0.27 **	−41.12 ± 0.46	−48.60 ± 1.29 ***	−45.00 ± 0.85 ***
pH	In	7.17 ± 0.01	7.25 ± 0.01	7.31 ± 0.01	7.24 ± 0.00	7.30 ± 0.01	7.36 ± 0.00
	2 y	3.21 ± 0.02 ***	3.74 ± 0.01 ***	3.26 ± 0.02 ***	3.63 ± 0.02 ***	3.21 ± 0.01 ***	3.46 ± 0.01 ***
Conductivity (µS/cm)	In	169.43 ± 1.46	179.20 ± 1.30	177.03 ± 0.12	180.07 ± 0.25	180.50 ± 0.40	189.20 ± 0.17
	2 y	305.67 ± 1.15 ***	216.67 ± 0.55 ***	299.00 ± 1.00 ***	242.33 ± 1.53 ***	319.67 ± 2.08 ***	290.33 ± 3.06 ***

F51—placebo and CF51—curcumin loaded with 0.1%; F53—placebo and CF53—curcumin loaded with 0.3%; F26—placebo and CF56—curcumin loaded with 0.6% of the PEG-PL. Values are shown as means ± sd (*n* = 3); *, **, and ***, *p* < 0.05, *p* < 0.01, and *p* < 0.001 compared to the initially measured values.

**Table 9 pharmaceutics-14-01666-t009:** DPPH assay obtained IC50 values.

Formulations	IC50 (mg/mL)	
	Initial	2 Years
CS	0.0805 ± 0.0022	0.0737 ± 0.0029 *
CS21	0.0779 ± 0.0051	0.0696 ± 0.0010
CS51	0.0752 ± 0.0009	0.0695 ± 0.0010 **
CF	0.0838 ± 0.0005	0.0767 ± 0.0011 ***
CF21	0.0867 ± 0.0040	0.0760 ± 0.0006 *
CF51	0.0873 ± 0.0026	0.0787 ± 0.0015 **

Values are shown as means ± sd (*n* = 3); *, **, and ***, *p* < 0.05, *p* < 0.01, and *p* < 0.001 compared to the initially measured values.

## Data Availability

Not applicable.

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
