# Peer review of "The Impact of the Oil Phase Selection on Physicochemical Properties, Long-Term Stability, In Vitro Performance and Injectability of Curcumin-Loaded PEGylated Nanoemulsions"

_pharmaceutics, 2022, doi:10.3390/pharmaceutics14081666_

Round 1

Reviewer 1 Report

The work tried to fabricate curcumin-loaded nanoemulsions, and the impact of oil phase and PEGylated phopholipids was studied. Overall, the organization of experiments and data is not satisfied to the journal.

1. Title should be simple and clear. It can be impact of oil phase and PEGylated phospholipids on long term stability of curcumin-loaded nanoemulsions and  in vitro performance evaluation.

2. abstract, it should be clear about the composition of nanoemulsions.  do not use too many short names in the abstract and keywords.

3. The research progress of curcumin-loaded nanoemulsiosns should be reviewed to emphasize the novelty of the present work.

4. Ln 324-327, it said that curcumin had the solubility in fish oil is 0.7551 mg/ml (ln 324), and 1.4526 mg/ml in fish oil (ln 326). Ln 326, it should be in the fish oil mixture?

5. There are too many figures and tables. Please re-organize the results.  It is difficult to read Figure 4, 5, 12, 13.

6. why curcumin had the higher release rate in fish oil formulations than that in soybean oil ones?

7. What is the advantage by using PEG modified phospholipids?

Author Response

Please, see our replies in the attached document

Reviewer 2 Report

The authors have carried out the preparation and characterization of curcumin loaded nanoemulsion and employed Electron paramagnetic resonance spectroscopy (EPR) technique for understanding the impact of some factors affecting the nanoemulsion stabilization. The study seems interesting and has the potential to get published in the journal in question. The data analysis has bee carried out extensively and data presentation also seems appropriate. The manuscript is well written and can be of ample interest to the readers of journal and other researchers working in the pharmaceutics arena. There are several following concerns which need to be answered before the manuscript can be finally accepted for publication.   

1.     Instances of the English language grammatical errors as for example in the ethical statement, need to be carefully rectified (e.g., curcumin is a potent …in line 26 of abstract; “could be used employed in prospective” in line 42 of abstract;)

2.     Can the authors provide the exact source and percentage purity of all the chemicals used for the study under the materials section?

3.     The major or final and optimised particle size, zeta potential and polydispersity index values could be represented with he histograms obtained form the Malvern zeta sizer (Dynamic light scattering) instrument files.  

4.     Introduction section seems a bit brief or short and can be supplemented with more details of Electron paramagnetic resonance spectroscopy (EPR) and its advantages, DoE details etc.

5.     Did the authors perform transmission electron microscopy (TEM) analysis of the nanoemulsion? It would be better if authors can provide the TEM analysis for particle size and morphological characters.

6.     In figure 9 cumulative drug release studies, the bars for the standard deviation are missing in soybean oil (CS) group. Like wide the bars in CS21 and CS23 seems missing in figure 10. Similar is the case with CS53 and CS56 in figure 11.

7.     Did the authors carry out any of the in-vitro cell line studies for the antioxidant potential of the nanoemulsion?

8.     Can the authors supplement some of their data with in-vivo studies?

9.     More recent references can be cited in the results and discussion section.

Author Response

(The authors gave the same response as above.)

Reviewer 3 Report

Review comments on pharmaceutics-1815554: Design of experiments coupled with structural analysis in optimization of curcumin-loaded nanoemulsions: impact of oil phase and PEGylated phospholipids on long term stability and in vitro performance

1. The manuscript by Đoković et al. described the preparation and characterization of curcumin-loaded nanoemulsions. The authors evaluated the effects of the oil phase and PEGylated phospholipids on the nanoemulsions in terms of stability and in vitro performance. Although the authors presented a large amount of data, they seemed to be piled up as a technical report rather than a study. The authors should refine the data and make the results and discussion more concise.

2. The authors should highlight the contribution of this study. It is hard to see the importance of this study and how it can contribute to the field.

3. The authors should evaluate in-depth how the structure of PEGylated phospholipids affect the nanoemulsions.

4. The Abstract is quite confusing. Please consider rewriting it.

5. There are many previous studies on curcumin-loaded nanoemulsions. The authors should cite and introduce them in the Introduction. Also, in the Introduction, the author should clarify the novelty and contribution of this study.

6. Section 2.2.1- Solubility of curcumin in oils: oils can affect the curcumin absorbance at 425 nm. The authors should verify it by measuring the absorbance of diluted oils (in MeOH) at 425 nm.

7. Line 246: the concentrations of curcumin were unusual. Generally, one may prefer concentrations like 0.002, 0.004, 0.006 mg/mL. Why were these concentrations selected?

8. Tables 4, 5, 6, 7, 8, and 9: significant symbols were for “compared to the initially measured values”. Thus, the symbols should be placed next to “2 y” values.

9. Conclusion: this section should be shortened. All discussion parts should be placed in the Discussion section. The authors should briefly conclude what can be withdrawn from this study.

Author Response

(The authors gave the same response as above.)

Round 2

Reviewer 1 Report

Author has carefully revised the paper. Now the presentation of the paper is clearer. I would suggest the publication in the journal.

Reviewer 2 Report

The revision and improvement of the manuscript seem satisfactory and the manuscript can be accepted for publication.

Reviewer 3 Report

The manuscript was revised accordingly. However, there are still some issues to consider as follows.

1. The Abstract should be shortened for conciseness. The authors should clarify some abbreviations in the Abstract (PEG, NE, DPPH, and FRAP).

2. The absorbance of diluted oils (in MeOH) at 425 nm (attached in the response letter) can be included in Supplementary materials to support the method development.